# Extracellular Vesicles in Inner Ear Therapies—Pathophysiological, Manufacturing, and Clinical Considerations

**DOI:** 10.3390/jcm11247455

**Published:** 2022-12-15

**Authors:** Athanasia Warnecke, Hinrich Staecker, Eva Rohde, Mario Gimona, Anja Giesemann, Agnieszka J. Szczepek, Arianna Di Stadio, Ingeborg Hochmair, Thomas Lenarz

**Affiliations:** 1Department of Otolaryngology, Hannover Medical School, 30625 Hannover, Germany; 2Cluster of Excellence of the German Research Foundation (DFG; “Deutsche Forschungsgemeinschaft”) “Hearing4all”, 30625 Hannover, Germany; 3Department of Otolaryngology Head and Neck Surgery, University of Kansas School of Medicine, Rainbow Blvd., Kansas City, KS 66160, USA; 4GMP Unit, Spinal Cord Injury & Tissue Regeneration Centre Salzburg (SCI-TReCS), Paracelsus Medical University, 5020 Salzburg, Austria; 5Transfer Centre for Extracellular Vesicle Theralytic Technologies (EV-TT), 5020 Salzburg, Austria; 6Department of Transfusion Medicine, University Hospital, Salzburger Landeskliniken GesmbH (SALK) Paracelsus Medical University, 5020 Salzburg, Austria; 7Research Program “Nanovesicular Therapies”, Paracelsus Medical University, 5020 Salzburg, Austria; 8Department of Diagnostic and Interventional Neuroradiology, Hannover Medical School, Carl-Neuberg-Str. 1, 30625 Hannover, Germany; 9Department of Otorhinolaryngology, Head and Neck Surgery, Charité-Universitätsmedizin Berlin, 10117 Berlin, Germany; 10Faculty of Medicine and Health Sciences, University of Zielona Gora, 65-046 Zielona Gora, Poland; 11Department GF Ingrassia, University of Catania, 95124 Catania, Italy; 12MED-EL Medical Electronics, Fürstenweg 77a, 6020 Innsbruck, Austria

**Keywords:** extracellular vesicles, exosomes, multipotent mesenchymal stromal cells, umbilical cord, inflammation, oxidative stress, cochlea, hearing loss, cochlear implantation

## Abstract

(1) Background: Sensorineural hearing loss is a common and debilitating condition. To date, comprehensive pharmacologic interventions are not available. The complex and diverse molecular pathology that underlies hearing loss may limit our ability to intervene with small molecules. The current review foccusses on the potential for the use of extracellular vesicles in neurotology. (2) Methods: Narrative literature review. (3) Results: Extracellular vesicles provide an opportunity to modulate a wide range of pathologic and physiologic pathways and can be manufactured under GMP conditions allowing for their application in the human inner ear. The role of inflammation in hearing loss with a focus on cochlear implantation is shown. How extracellular vesicles may provide a therapeutic option for complex inflammatory disorders of the inner ear is discussed. Additionally, manufacturing and regulatory issues that need to be addressed to develop EVs as advanced therapy medicinal product for use in the inner ear are outlined. (4) Conclusion: Given the complexities of inner ear injury, novel therapeutics such as extracellular vesicles could provide a means to modulate inflammation, stress pathways and apoptosis in the inner ear.

## 1. Introduction

More than 450 million people worldwide suffer from hearing loss [1]. According to the World Health Organization, one in twenty individuals is affected by hearing loss [2] and at risk of developing associated co-morbidities such as depression and dementia [3]. Although the importance of treating hearing loss to prevent these conditions has been demonstrated [4], pharmacological treatments are still unavailable, leaving most patients not adequately treated. Because some forms of hearing loss are age-related, the increase in the elderly population affects the prevalence of this symptom. For this reason, identifying effective treatments is fundamental also to reducing co-morbidities, i.e., cognitive decline [5].

A comprehensive understanding of the molecular pathophysiology of different inner ear diseases is lacking. Understanding the exact molecular mechanisms that lead to damage to the inner ear and induce hearing loss would accelerate the development of effective precision treatments. With innovations in molecular biology, some cellular and molecular changes related to hearing loss are elucidated and suggest the involvement of multiple diverse injury and stress pathways which are unlikely to be completely treated using standard pharmacological therapy. The current review focuses on the molecular changes associated with inner ear damage and hearing loss, which may be treatable using extracellular vesicles to directly target the molecular changes related to aging and inner ear damage. Finally, manufacturing and clinical considerations associated with introducing EV-related treatment strategies in the clinic will be discussed. This narrative review is based on searches performed in PubMed and ClinicalTrials.gov.

## 2. Hearing Loss

The causes for hearing loss are diverse and include genetic predisposition, ototoxic agents, and environmental factors such as noise and aging. Depending on the severity of the insult, different grades of damage are observed within the cochlea leading to different phenotypes of hearing loss and ranging from reversible to permanent: starting from molecular damage such as induced by excitotoxicity (e.g., loss of ribbon synapses) to more pronounced cellular and structural degeneration (loss of connectivity from neurons to hair cells, loss of hair cells, spiral ganglion neurons and cells of the stria vascularis). Ultimately, the neuroepithelium degenerates, leaving a flat non-functional epithelium in the organ of Corti [6,7]. Different prosthetic devices are available for hearing restoration depending on the severity of the hearing loss. Patients with moderate hearing loss can be treated with hearing aids (HA) (simple sound amplification). Unfortunately, many patients refuse daily use of their HA due to a lack of benefit in many everyday listening situations [8,9]. Cochlear implantation for direct electrical stimulation of the auditory neurons is performed in cases with severe hearing loss across all frequencies. In recent years, the use of cochlear implantation has been expanded to patients with residual hearing (high-frequency hearing loss but preserved hearing for lower frequencies). In these cases, hybrid stimulation is provided via a hearing aid (that amplifies low-frequency information acoustically) in combination with a cochlear implant (that electrically stimulates the high-frequency regions) [10,11]. With this strategy, patients develop a superior speech understanding even in challenging listening situations [12,13]. However, a significant portion of these patients loses their residual hearing after cochlear implantation [14] limiting the long-term efficacy of hybrid stimulation. There is an unmet clinical need for effective treatment strategies to protect residual hearing. In patients with residual hearing, the apical regions responsible for processing acoustic stimuli in lower frequency have intact inner ear cytoarchitecture and preserved electrolyte homeostasis and blood supply despite a damaged basal region (the area processing high-frequency sound).

## 3. Inflammation in Hearing Loss

Acute inflammation is an adaptive response to combat invading pathogens or to repair damaged tissues. An acute inflammatory reaction is a highly coordinated process [15]. Resolution of inflammation is critical to restore organ homeostasis and to prevent ongoing inflammatory responses leading to tissue damage [15,16,17,18].

The inflammation arises as a common hallmark underlying hearing loss [18]. Cellular stress leads to producing reactive oxygen species (ROS) [19,20,21], activating pro-survival pathways and increasing endogenous antioxidant molecules to maintain redox homeostasis [22]. Excessive ROS can cause either damage to the tissue directly by oxidation and reduction of cochlear blood flow [23] or indirectly by up-regulating pro-inflammatory cytokines, e.g., interleukin-6 (IL-6) and tumor necrosis factor-alpha (TNFα) [23]. These cytokines, in turn, induce apoptosis or necrosis [23]. Endogenous pathways that initiate the inflammation depend mainly on ROS [24]. Accumulation of ROS precedes any morphological sign of damage [25]. It spreads from the basal to the apical part of the cochlea [26], subsequently damaging initially unaffected parts as has been shown, e.g., for noise-exposure, ototoxicity, aging, Ménière’s disease, and in cochlear implantation [22,23,25,26,27,28,29,30,31]. In fact, cochlear implantation can add additional damage to an already diseased organ through the surgical opening of the cochlea and insertion of a foreign body (the electrode). Specifically, foreign body insertion can lead to a phagocytic reflex in macrophages and activation of the NACHT-, leucine-rich repeat-and pyrin-domain-containing protein (NALP3 or NLRP3) inflammasome, which is a sensor of inflammation [24]. Thus, we create a wound not only at the opening site but also within the cochlea that elicits a trauma reaction with inflammation [32,33] leading to tissue remodeling and fibrosis [34,35]. Postmortem analyses of human temporal bones have confirmed that implantation trauma can result in severe damage of the cochlear structural and ultrastructural components [34,35,36,37] and loss of residual hearing [30,32,38].

The impact of the molecular changes associated with electrode insertion trauma has been recently reviewed in detail [39]. Injury to the stria vascularis in the lateral cochlear wall can be caused when inserting stiff straight electrode arrays, and this injury leads to an increase in ROS and induces inflammation. Consequently, the vascularization of the stria vascularis decreases [39], as has been shown in an animal model of cochlear implantation with residual hearing [40]. The fibrocytes that reside in the cochlear lateral wall are thought to mediate inflammation after noise exposure by the release of pro-inflammatory factors [41,42] such as interleukin (IL)-1β, TNF-α, inducible nitric oxide (NO), monocyte chemoattractant protein-1 (MCP-1), macrophage inflammatory protein (MIP), intercellular adhesion molecule-1 (ICAM-1), nuclear factor kappa B [43], and vascular endothelial growth factor (VEGF) [44]. Resident macrophages are activated and up-regulated by inner ear damage, as has been shown for noise exposure or aminoglycoside toxicity [44,45]. Indeed, the inhibition of macrophages resulted in protection against aminoglycoside toxicity [46].

Acute inflammation upon an insult may also be considered a necessary host reaction to restore cochlear homeostasis and glial-neuronal interactions. For example, neuroactive molecules such as TNFα and IL-1β are also released in patients with sudden sensorineural hearing loss, and increased levels can be measured in their peripheral blood [47,48]. The release of such cytokines might be essential to induce inflammation, followed by a timely resolution in order to prevent auditory dysfunction [18]. Indeed, maladaptive inflammation is acknowledged as a cause of neurodegenerative diseases, asthma, allergy, diabetes, fibrosis, cardiovascular disease, and metabolic disorders [17]. Four phases characterize inflammatory responses, starting with initiation, transition, resolution, and finally, restoration of homeostasis [15,16,17,18]. Others classify inflammation by the onset, resolution, and post-resolution phases [49]. Inflammation is initiated by recognizing pathogen- and/or damage-associated molecular patterns [17]. This phase is characterized by a coordinated delivery of blood components to the site of infection or injury [24] mediated by increased blood flow, capillary dilatation, leukocyte infiltration, as well as production of pro-inflammatory mediators, including cytokines, chemokines, vasoactive amines, and eicosanoids [49,50]. The release of pro-resolving chemical mediators characterizes the resolution phase, which aims to eliminate the inflammatory trigger, the clearance of immune cells from the inflamed sites, the reduction in extravasation of cells from the circulation, and the change from pro-inflammatory to pro-tissue repair signaling pathways [24].

On a molecular level, connexin43 channels and hemichannels have been demonstrated to amplify and perpetuate inflammation [50]. Inflammation can be initiated and maintained via adenosine triphosphate (ATP) released by connexin hemichannels [50]. In the cochlea, ATP release has also been shown to occur via connexin 26 and 30 hemichannels [51,52]. An increase in ATP concentration has been demonstrated in the cochlea, particularly after noise trauma, and may impair cochlear function [53]. Connexin hemichannels are located on the non-junctional cell surface and are not involved in forming gap junctions; they can open due to depolarization, hyperpolarization, or metabolic stress and release increased ATP [52]. The ribbon synapse protein CtBP can detect NAD+ and NADH levels and is therefore considered a metabolic biosensor [54,55]. Moreover, changes in the NAD(H) redox status can affect the formation of CtBP and, thus, that of the synapses [54,56]. Mitochondria are localized near the presynaptic ribbons in hair cells [54]. Mitochondria are the energy producers of the cell and the primary source of cellular ATP. As by-products, reactive oxygen species are also produced and released in the mitochondria while providing ATP. ATP is formed after the oxidation of carbohydrates, lipids, and proteins as energy storage. Other metabolites such as inositol 1,4,5 trisphosphate (IP3), glutamate, lactate, and D-serine are also released. ATP activates the NLRP3 inflammasome [50]. The exact mechanisms leading to a prolonged or dysregulated resolution of inflammation in the cochlea and other organ systems are unknown.

The late enhancement on magnetic resonance imaging is a potential sign of cochlear inflammation that can be visualized. Intralabyrinthine enhancement has been a diagnostic finding in T1-weighted magnetic resonance imaging (MRI) for many years. This enhancement usually refers to an enhancement a few minutes after intravenous (i.v.) contrast administration and is found in the acute stage of labyrinthitis ossificans and intracochlear schwannoma. Other causes are rarely encountered. In the last decade, the so-called “hydrops”-imaging has been performed to visualize an enlargement of the membranous spaces of the inner ear [57]. It is performed 4 to 6 h after i.v. contrast administration. A certain amount of contrast that gets into the perilymphatic space distinguishes the scala vestibuli and the scala tympani from the scala media on heavily T2-weighted 3D Flair images. At the same time, substantial enhancement was associated with several conditions. In patients with vestibular neuritis an enhancement can be seen regularly [58]. The same is true for about 20% of the patients with otosclerosis [59]. Single cases were attributed to a herpes infection. In many other cases, the causes remain unknown, but autoimmune diseases such as Cogan syndrome or accompanying inflammation in cases with known scleroderma or rheumatoid arthritis were discussed. Further studies have to evaluate the possible causes for the impaired blood–labyrinth barrier in detail—but late intralabyrinthine enhancement in MRI is a promising tool to detect infectious and inflammatory conditions of the cochlea and the vestibule.

## 4. Electrotoxicity

Damage of synaptic contacts due to electrical stimulation might be another reason for loss of residual hearing after cochlear implantation [60,61]. Indeed, chronic electrical stimulation leads to hearing loss, as has been demonstrated in animal models and the loss of hearing seems to be histologically attributed to the loss of synapses [40,60]. In the organ of Corti explants, a reduced density of ribbon synapses, an increase in free reactive oxygen species, and morphological changes in stereocilia bundles were observed under electrical stimulation [62,63]. The treatment with dexamethasone [64] could partially prevent the damage induced by experimental electrical stimulation. The exact molecular mechanisms and the conditions under which electrical stimulation may damage hair cells, and spiral ganglion neurons are unknown. In vitro experiments using charge-balanced biphasic electrical stimulation showed a reduced neurite outgrowth from the spiral ganglion and a reduced density in Schwann cells, possibly due to calcium influx through multiple types of voltage-gated calcium channels [65]. Although calcium overload and oxidative stress are assumed to be the leading cause of spiral ganglion neuron degeneration, decreased intracellular levels of calcium have been observed in neurons damaged by electrical stimulation [66]. Indeed, neuronal death has been observed under low and high calcium levels, and an imbalance in intracellular calcium homeostasis might increase the vulnerability of spiral ganglion neurons under electrical stimulation [66]. A retrospective analysis of patients enrolled in the multicenter Hybrid S8 trial who were treated with a Nucleus Contour Advance perimodiolar standard length electrode array or a Nucleus 422 Slim Straight electrode array showed an acceleration of hearing loss after activation of the device [61]. In addition, high charge exposure with the same devices also led to accelerated loss of residual hearing [61]. To our knowledge, this is the first and only report on humans showing a correlation between electrical activation and hearing loss. The authors also corroborated their results experimentally on the organ of Corti explants showing that high voltage ES damaged afferent nerve fibers [61]. Whether and under which stimulation parameter electrotoxicity can be mediated via all cochlear implants in humans is unknown hitherto.

## 5. Extracellular Vesicles

The term “extracellular vesicles” refers to a heterogeneous set of cell-derived membranous nano- and microvesicles that can be harvested for diagnostic and therapeutic purposes from the secretome of cells and any body fluid. Small EVs display a diameter ranging from 50–150 nm and are also called exosomes. Larger microvesicles up to 1000 nm and apoptotic bodies have also been identified. Potentially all pro- and eukaryotic cells release EVs into their secretome. There are two ways for the biogenesis of extracellular vesicles. By the inward budding of the plasma membrane or the trans-Golgi network, early endosomes are formed and mature into late endosomes to finally become multivesicular endosomal bodies (MVBs) [67,68]. After trafficking to the cell membrane, a fusion of MBV with the plasma membrane leads to the release of exosomes into the extracellular space. Another mechanism is the plasma membrane rearrangement leading to the budding of microvesicles from the cell membrane [67,68]. Through various isolation techniques, EVs can be enriched in a vesicular secretome fraction [69,70]. Depending on their source, EVs can be involved in physiological or pathophysiological processes. For example, exosomes derived from malignant cells can promote tumor growth, local invasion, and distant metastasis [71,72,73,74]. During pregnancy, placental exosomes are involved in the regulation and progression of a normal pregnancy and pathological conditions that can arise during pregnancy [75]. Depending on their source, EVs exert differential effects. For example, anti-apoptotic [76,77], anti-fibrotic [78,79,80], pro-angiogenic [81,82], and immunomodulatory [83,84,85,86,87,88,89,90] effects are mediated by EVs derived from naïve umbilical cord mesenchymal stromal cells (UC-MSC).

## 6. Extracellular Vesicles as Anti-Inflammatory and Anti-Oxidative Treatment

Growth factors such as the neurotrophins brain-derived neurotrophic factor (BDNF) and neurotrophin-3 (NT-3) are essential regulators in embryonic development [91,92] and in the maintenance of the adult auditory system [93,94,95]. In addition, individual neurotrophins have also been shown to exert immunomodulatory effects [96,97]. However, in terms of potency, a cocktail of various growth factors was more effective in the protection of neuronal survival than single factors [98,99,100]. For the delivery of human neuroprotective factor cocktails to the inner ear, we have investigated cell-based approaches such as platelet-rich plasma [101] or autologous mononuclear cells derived from human bone marrow [102] as potent regulators of inflammation and oxidative stress and as a source of a balanced composition of endogenous neuroprotective factors. We assume that extracellular vesicles (EVs) are the main contributors to neuroprotection and regulation of inflammation when using cell-based approaches.

Extracellular vesicles mainly target immune-competent cells such as T cells, B cells, and natural killer cells. Like their parental cells, MSC-EVs can impede the maturation of dendritic cells and modify their function, thereby suppressing antigen uptake [103]. In addition, the cytokine production profile of dendritic cells was changed from pro-inflammatory to immunoregulatory after exposure to MSC-EVs [103]. A phenotype change from pro-inflammatory to immunoregulatory was also observed in macrophages after exposure to EVs [104]. Specific microRNAs involved in the development and maturation of dendritic cells are highly enriched in MSC-EVs and might account for the anti-inflammatory and immunomodulating effects [103]. Chronic unresolved inflammation with an accumulation of pro-inflammatory macrophages is the hallmark of metabolic diseases, including diabetes. The release of pro-inflammatory cytokines such as tumor necrosis factor-alpha (TNF-α) and interleukin-1-beta (IL-1β) are critical drivers for the maintenance of inflammation and the induction of insulin resistance. Indeed, macrophages exposed to high glucose levels and oxidative stress release EVs with an altered microRNA profile that can induce atherosclerosis [105]. Th2 cytokines such as IL-4 and IL-13 polarize macrophages towards an anti-inflammatory phenotype [106,107,108] with associated changes in their EV-cargo [105,109] and protective effects against cardiometabolic disease [109].

Excessive release of pro-inflammatory cytokines leading to an overwhelming systemic inflammation is the main cause of severe organ damage and failure in sepsis developed after infectious diseases [110,111]. Mesenchymal stromal cells have an immunomodulatory effect and could be ideal candidates for modulating the deregulated immune response in sepsis [112,113,114]. When primed with IL-1β the immunomodulatory efficacy of MSC could be increased [87]. A significant upregulation of miR-146a expression and packaging within exosomes seems to be the mechanism by which priming with IL-1β increases the potency of MSC to ameliorate the symptoms associated with sepsis and prevent organ injury [87]. Additionally, MSC and MSC-EVs mediate a transition of macrophages from a pro-inflammatory to an immunomodulatory phenotype that is beneficial for recovery from sepsis [87]. Derived from macrophages with an anti-inflammatory phenotype, EVs have been shown efficient in reducing excessive cytokine release and oxidative stress leading to multiple organ damage in mice challenged with bacterial endotoxins [115]. Specifically, the release of pro-inflammatory cytokines TNF-α and IL-6 is reduced [115]. That is important since TNF-α is the primary activator of the pro-inflammatory NF-κB pathway amplifying the inflammatory cascade leading to lethal toxic shock [115,116,117].

The cytokines TNF-α and IL-6 are also involved in severe steroid-resistant asthma and exacerbate airway inflammation and lung tissue damage [118,119]. Indeed, MSC-EVs were able to ameliorate inflammation via the NF-κB and PI3K/AKT signaling pathways by reducing the expression of the TNF-receptor-associated factor 1 (TRAF1) [83]. TRAF1 is involved in regulating inflammation and apoptosis and is known to activate NF-κB [83]. Thus, MSC-EVs may present an ideal targeted therapeutic for severe steroid-resistant asthma.

Suppression of lymphocyte proliferation [86], specifically B cells and natural killer cells, was mediated by anti-inflammatory factors such as indoleamine 2 3-dioxygenase (IDO) [86]. In inflammatory skin diseases such as psoriasis, IL-17A, and its upstream regulator IL-23 are two critical molecules involved in the pathogenesis and thus present key targets for targeted treatment [120]. Extracellular vesicles derived from UC-MSC promote a shift from a Th1 or Th17 into a Treg phenotype, thus, exhibiting the ability to silence Th17 signaling in patients with psoriasis [85]. Multiple sclerosis (MS) is characterized by the proliferation of conventional T cells and their differentiation into an autoreactive phenotype in response to self-antigens [121]. Treatment with EVs reduced IFN-γ and IL-17 release from lymphocytes derived from patients with MS [84].

The emerging role of EVs in promoting intracellular defense against oxidative stress by inhibiting the formation of excess ROS, thereby improving mitochondrial performance, has been discussed in a recent review article [122]. Thus, EVs may be effective therapeutics to treat oxidative stress. For example, EVs derived from patients’ blood after myocardial infarction have an antioxidative effect on the endothelium [123]. In response to heat shock, EVs can protect cultured neurons from oxidative stress [124]. Derived from hypoxia preconditioned MSC, EVs have been shown to protect cardiomyocytes from apoptosis by targeting a protein [125] (thioredoxin-interacting protein) involved in oxidative stress responses [126]. In diabetic cardiomyopathy, the platelet inhibitor ticagrelor modulates the cargo of EVs to enhance the suppression of cellular stress and ROS production [127]. Several heat shock proteins are involved in oxidative stress, either protective or damaging. The heat shock protein B8 (HSPB8) is involved in removing cytotoxic proteins, thereby facilitating autophagy and reducing oxidative stress. Endogenous HSPB8 mRNA expression is increased by the uptake of EVs derived from oligodendrocytes [128]. However, the protective effect of EVs from oxidative stress is not inherent and depends on the source and the content of EVs. The transcription factor nuclear factor (erythroid-derived 2)-like 2 (Nrf2) regulates the expression of antioxidant and anti-inflammatory proteins in response to oxidative stress [129]. However, the effects of EVs are not inherent and depend on the source and the content of EVs. As a result, EVs can contribute to the healing process but also induce inflammation and disease. The transcription factor nuclear factor (erythroid-derived 2)-like 2 (Nrf2) regulates the expression of antioxidant and anti-inflammatory proteins in response to oxidative stress [129]. Cardiac-derived EVs isolated from the circulation of rats and patients suffering from chronic heart failure have a miRNA profile associated with inhibition of the Nrf2/anti-oxidant signaling pathway, thereby contributing not only to the heart disease but also to the neuroinflammation and redox imbalance at distant sites such as the brain [129]. Altering the cargo of EVs or inhibiting the responsible miRNA may present a therapeutic strategy for treating chronic heart failure. 

## 7. Extracellular Vesicles as a Novel Therapeutic in Neurotology

There is a lack of specific inner ear therapeutics for the protection and regeneration of cochlear cells alongside an antioxidative and anti-inflammatory/immunomodulatory treatment.

The first report about the release of EVs from rats’ primary cultured inner ear cells was published by Wong et al. in 2018 [130]. The authors also showed that exosomes could potentially be used as a biomarker reflecting the state of the inner ear [130]. For example, there was a change in concentration and proteomic cargo upon induction of ototoxic stress by exposure to cisplatin or gentamycin [130]. The authors state that the isolated exosomes may be derived from the organ of Corti [130]. Furthermore, EVs isolated from a murine auditory cell line (HEI-OC1) showed a distinct protein and surface marker profile of HEI-OC1-EVs [131] when compared to the EVs content listed in ExoCarta, a manually curated database of exosomal proteins, RNA, and lipids. Moreover, the authors stated that HEI-OC1-EVs could be loaded with anti-inflammatory drugs and pro-resolving mediators and used as drug nanocarriers [131,132].

For the first time, using human perilymph, we could isolate EVs carrying hair cell-specific proteins [133], demonstrating not only the presence of EVs in the perilymph but also suggesting hair cells as a potential source of the exosomes. Cochlear tissues derived from different postnatal developmental stages were analyzed for EVs’ presence and content. Indeed, EVs had a specific miRNA and proteome profile related to the development of the inner ear and auditory nervous system [134]. Although that study suggests that EVs may be involved in inner ear development, the presence and role of exosomes in the adult inner ear are still unclear. As in other organ systems, EVs might contribute to physiological and pathophysiological processes in the adult inner ear. For example, it has been shown that EVs derived from human vestibular schwannoma (VS) cell culture can exert differential effects on hair cells and auditory neurons [135]. Exosomes derived from patients diagnosed with VS and hearing impairment were more likely to damage hair cells and auditory neurons than exosomes derived from patients with VS and good hearing [135].

The protective role of exosomes in otology has been demonstrated in several preclinical studies [136,137,138,139,140,141,142,143]. Breglio et al. were the first to demonstrate in 2020 that utricle-derived exosomes are mediators of heat stress response in the inner ear and can protect hair cells against aminoglycoside-induced death [142]. We were the first to also demonstrate in 2020 that umbilical cord-derived MSC-EVs exerted immunomodulatory activity on T cells and microglial cells and significantly improved spiral ganglion neurons’ survival in vitro [143]. We furthermore could show that MSC-EVs contain BDNF and that local post-traumatic application of MSC-EVs to the cochlea attenuated hearing loss and protected auditory hair cells from noise-induced trauma in vivo [143]. Our results were corroborated by Tsai et al., who also showed later that post-traumatic administration of UC-MSC-EVs significantly improved hearing loss and rescued the loss of cochlear hair cells in mice receiving chronic cisplatin injection [140]. Preconditioning of MSC may increase the therapeutic efficacy of thereof derived EVs. For example, hypoxia-preconditioned MSC-EVs have an up-regulated expression of HIF-1α, leading to increased potency in the treatment of cisplatin-induced ototoxicity [139]. Heat shock-preconditioned MSCs release EVs with an increased HSP70 content and an increased efficacy against cisplatin-induced hair cell loss [136]. HSP70 might be one of the mediators of EV-related protective effects. Interestingly, the co-culture of MSC and cochlear explants led to an increase in HSP70 content in EVs and was able to protect explants from cisplatin-induced toxicity [139]. Also, exosomal HSP70 treatment decreased the concentration of pro-inflammatory cytokines IL-1β, IL-6, and TNF-α and increased the concentrations of anti-inflammatory cytokine IL-10 in cisplatin-exposed mice inner ears [136]. Neural progenitor cells also release EVs. Specifically, EVs derived from miRNA-21-overexpressing progenitors, showed increased anti-inflammatory activity and prevented hearing loss after ischemia-reperfusion injury [137]. Interestingly, the effects were also mediated by a decrease of IL-1β, IL-6, TNF-α, and an increase of IL-10 [137]. Exosomes derived from spiral ganglion progenitor cells isolated from neonatal mice cochlear explants were able to protect the hearing threshold in mice exposed to ischemia-reperfusion injury [141].

The feasibility of applying allogeneic human MSC-EVs into the inner ear was recently demonstrated in the first report of EV application into the scala tympani during cochlear implantation [144]. The goal of cochlear implantation would be to limit the potential damage induced in the inner ear by a foreign body reaction, inflammation caused by the surgical opening of the cochlea, and insertion of the electrode array, and to protect residual hearing for improved implant performance. Other application fields for EVs in neurotology are the increase of efficacy of viral vector gene therapy approaches. It has been shown that exosomes applied in conjunction with AAV could rescue hearing in a mouse model of hereditary deafness [145]. Indeed, robust transduction with AAV was also observed in other organ systems, such as the eye or the nervous systems, by combining gene therapy with exosome treatment [146].

For a better overview on the studies demonstrating inner ear protection by EVs, Table 1 summarizes all to our knowledge currently published reports. Based on the potent anti-inflammatory efficacy of EVs and to the lack of specific treatment alternatives, treatment of autoimmune-mediated hearing loss, Meniere’s disease, and insertion of trauma-mediated immune responses could present potential clinical applications. To our knowledge, there are no clinical trials investigating the effect of EVs in the inner ear.

## 8. Manufacturing Considerations

EV-based therapeutics derived from genetically unmodified mesenchymal stromal cells are considered to belong to the pharmaceutical category of biological medicinal products in Europe, the United States of Amerika, Australia, and Japan [147]. A “biological medicine is a medicine that contains one or more active substances made by or derived from a biological cell” [147]. According to the European Medicines Agency (EMA, Amsterdam), medicines for human use that are based on genes, tissues, or cells are classified as advanced therapy medicinal products (ATMPs) [148]. “They offer ground breaking new opportunities for the treatment of disease and injury” [148]. The inherent heterogeneity and biological or technological complexity hamper the identification of the therapeutically active component or components in EV formulations and the mode of action. The latter depends on the parental cell type, handling and culture conditions, and materials or medical devices used for EV isolation and administration [149]. When considering translating the use of EVs towards clinical application, several considerations concerning their manufacturing are mandatory.

Harmonized guidelines are available, and the International Council for Harmonisation of Technical Requirements for Pharmaceuticals for Human Use connects regulatory authorities with researchers and biotechnology companies to discuss and regulate scientific and technical aspects of drug registration globally [149]. The goal is to ensure safety, effectiveness, and high quality for the manufacturing of ATMPs.

Product specifications related to purity, identity, quantity, potency, and sterility need to be defined according to pharmaceutical manufacturing regulations. Since EVs include a wide variety of membrane-bounded vesicles, exosomes are restricted by size and surface markers. Since part of the EV-based secretome includes soluble molecules such as proteins, lipids, and extracellular RNA species either from tissue or from in vitro expanded cell cultures, several factors may influence the composition of EVs. Electron microscopy data show that even highly purified EV preparations for analytical purposes contain co-purifying components [150]. Segregation of EVs from co-purifying components during large-scale clinical manufacturing will not be possible entirely, so an increasing fraction of secretome components will be present in the final preparation [149]. Thus, there is heterogeneity and complexity of secretome-based preparations, and an attempt to find a terminology that embraces all biological components and therapeutic aspects without eliminating the central claim is problematic. From a cell biology standpoint and not limiting the definition to a purely proteomic view, the secretome can be seen as the totality of organic molecules and inorganic elements secreted by cells into the extracellular space, either in a soluble or packaged form. Although the EVs’ manufacturing and enrichment process eliminates a large portion of soluble proteome components, the co-purifying fraction is still challenging to determine. Thus, the manufacturing strategy aims to enrich (and not necessarily purify) membrane-bounded vesicular structures [149]. The product may then be considered as a vesicular secretome fraction. This terminology can also accommodate fractions containing co-purifying soluble serum components in those cases where the manufacturing process requires the use of (vesicle-depleted) serum. Nevertheless, and fully considering the above, we will adhere to the EV terminology for simplicity. The product is a biological containing cell-derived EVs. The cell source for the production of EVs is manifold. Multipotent mesenchymal stromal cells-derived EVs might be favorable for a rapid clinical translation. From 2015–2021, 416 clinical trials comprising MSCs were registered [151]. In mid-2021, 1014 MSC-based clinical trials were registered in the ClinicalTrials.gov database either as completed or in progress [152]. A recent search in ClinicalTrials.gov (using the terms ‘application’ and ‘human umbilical cord-derived MSC’) showed that by October 2022, a total of 135 trials investigated the application of human umbilical cord-derived MSC in many diseases affecting adults and infants. Thus, abundant clinical data are available on the safety and efficacy of MSCs, specifically of UC-MSC.

The allogeneic human UC-MSCs grown in a fibrinogen-depleted growth medium containing pooled human platelet lysate (pHPL) serve as an alternative source for xenogeneic growth factors. The tissue of human origin must be collected following the Helsinki Declaration after the written informed consent of adult donors. A Master Cell Bank needs to be generated to provide a pool of producer cells for EVs. Since the final product should be a ready-to-inject solution, the sterilized product needs to be dissolved, e.g., in Ringer’s lactate, and filled at a defined dose in aseptic glass vials. The use of EV-based therapeutics instead of the cells themselves has several advantages: the possibility of filter sterilization of the final product immediately prior to aseptic filling and the stability after freeze–thaw cycles, as well as flexibility in the choice of storage buffers. However, compared to the cells as a product, EV manufacturing includes additional steps such as isolation, enrichment, and characterization of vesicles [149,153].

## 9. Regulatory Affairs

National competent authorities are responsible for approving clinical trials in the respective country and providing scientific and regulatory advice on drug development and the planning and conduct of clinical trials. Regulatory advice and authorization for drugs and ATMPs for human use are granted in Germany by the Paul Ehrlich Institute (PEI) or the Federal Institute for Drugs and Medical Devices (BfArM). When granted, European marketing authorizations are coordinated by the EMA and are valid for the entire European Union (EU).

The biological medicinal product must be manufactured under GMP-compliant conditions, and the guidelines apply to the whole manufacturing chain. Pharmaceutical production includes harvesting the tissue, isolation of the parental cells, culture environment, cultivation system, culture medium, isolation and purification of EVs, fill and finish as well as storage of the final product. According to the requirements of a state-of-the-art pharmaceutical quality management system, the quality and therapeutic activity of the EV-based novel product have to be confirmed by defined release testing. Biodistribution, bioavailability, cytotoxicity, and pharmacokinetics are cornerstones in the non-clinical development of a biological therapy toward an investigational medicinal product. Overall low toxicity of EVs has been demonstrated in several phase I clinical trials [154,155,156,157]. Labeling of EVs is mandatory for biodistribution studies if in vivo tracking of unlabeled EVs is not feasible in the relevant disease-specific animal models. Besides clinically approved radioisotopes, superparamagnetic particle loading is an alternative to investigating the distribution of EVs after application [149].

Validated in vitro and in vivo potency assays are necessary to systematically evaluate the expected biological activity and/or therapeutic potency in adequate models [149]. Depending on the disease and envisaged clinical application, potency assays, proof-of-concept, and application modes need thorough consideration and investigation. Of course, information about the therapeutically active substance of such complex novel biological therapies and the mode of action would help to accelerate translation to the clinic. Unfortunately, the molecules responsible for efficacy in the inner ear and EV mode of action are incompletely understood. Before the completion of the phase 2 clinical trial, there is no necessity to fully characterize and identify the active substance or to provide a detailed concept about the mode of action of an investigational medicinal product [147,158]. Outcome measures need not only be defined for planned clinical trials but also included as endpoints in preclinical experiments for an improved transfer and comparability. To this aim, an early and repeatedly seeking of advice from the responsible national and international authorities is highly recommended.

Partnerships between approving authorities, academia, and industry should be considered to pave and accelerate the road to the clinic for vesicle-based therapeutics. The International Society for Extracellular Vesicles ISEV has therefore built a “Task Force on Regulatory Affairs and Clinical Use of EV-based Therapeutics” to identify existing or develop novel applicable regulatory guidance [159]. The aim is to provide a safe and efficient evaluation of EVs in clinical trials for an evidence-based application of EV therapeutics for various pathological conditions.

## 10. Conclusions

The development of EV-based therapeutics in all fields, especially in neurotology, is on the rise and offers vast unexploited potential. Scalable and reproducible purification protocols are already available based on robust data and qualified potency assays in disease-specific in vitro and in vivo models. Following regulatory requirements and GMP compliance, high-quality clinical trials are rising in other medical fields. As such, EV products will be available for clinical use in the near future (Figure 1). Following the philosophy that “the developmental process is indeed the product”, each manufacturing step of EV-therapeutics needs to be standardized as much as possible. Thus, with the patient’s benefit in mind, joined efforts of regulatory authorities, academic experts, and biotechnological manufacturing teams can address the unmet clinical needs and will hopefully lead to a new era for treating acute and chronic diseases, especially in neurotology.

## Figures and Tables

**Figure 1 jcm-11-07455-f001:**
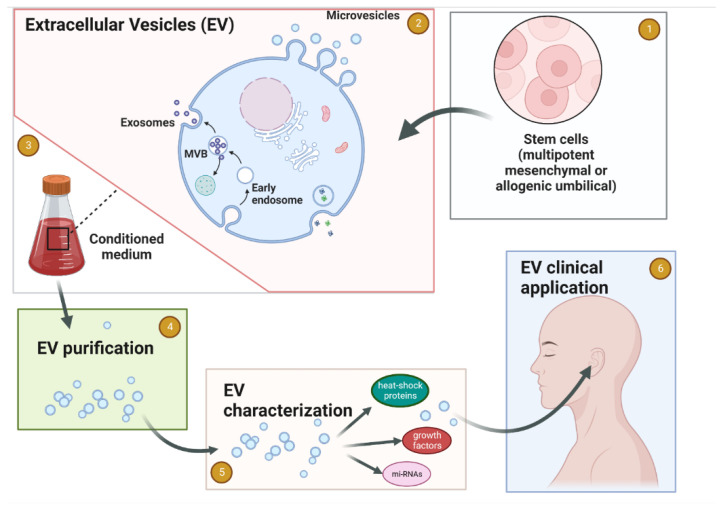
Schematic flow of EV production for medicinal use in Neurotology. (1) Establishing the cell source for EV. (2) Cells release the EV 3 to the tissue culture media. (3) Collection of the conditioned medium. (4) EVs purification. (5) EVs characterization. (6) Clinical application of purified and characterized EVs for the inner ear conditions. Created with BioRender.com.

**Table 1 jcm-11-07455-t001:** Summary of the studies demonstrating inner ear protection by EVs.

Authors	Title	Origin of EVs	Species	Journal and Year
Breglio, A.M. et al. [142]	Exosomes mediate sensory hair cells protection in the inner ear	Heat shocked utricles from mice	Mice	*J. Clin. Invest.* 2020
Lai, S.-W. et al. [97]	Exosomes derived from mouse inner ear stem cells attenuate gentamicin-induced ototoxicity in vitro through the miR-182-5p/FOXO3 axis	Inner ear stem cells from mice	Mice	*Mol. Neurobiol*. 2018
Warnecke, A. et al. [143]	Extracellular vesicles from human multipotent stromal cells protect against hearing loss after noise trauma in vivo	Human umbilical mesenchymal stromal cells	Mice	*Clin. Trans. Med.* 2020
Warnecke, A. et al. [144]	First-in-human intracochlear application of human stromal cell-derived extracellular vesicles	Human umbilical mesenchymal stromal cells	Human	*J. Extracell. Vesicles* 2021
Tsai, S.C.-S. et al. [140]	Umbilical Cord Mesenchymal Stromal Cell-Derived Exosomes Rescue the Loss of Outer Hair Cells and Repair Cochlear Damage in Cisplatin-Injected Mice	Umbilical mesenchymal stromal cells (presumably human; but not specified in publication)	Mice	*Int. J. Mol. Sci.* 2021
Yang, T. et al. [141]	Exosomes derived from cochlear spiral ganglion progenitor cells prevent cochlea damage from ischemia-reperfusion injury via inhibiting the inflammatory process	Spiral ganglion progenitor cells from mice	Mice	*Cell Tissue Res.* 2021
Jiang, P. et al. [134]	Characterization of the microRNA transcriptomes and proteomics of cochlear tissue-derived small extracellular vesicles from mice of different ages after birth	Cochlear tissue from mice	-	*Cell Mol. Life Sci.* 2022
Hao, F. et al. [137]	Exosomes Derived from microRNA-21 Overexpressing Neural Progenitor Cells Prevent Hearing Loss from Ischemia-Reperfusion Injury in Mice via Inhibiting the Inflammatory Process in the Cochlea	Neural progenitor cells from mice transfected with miR-21	Mice	*ACS Chem. Neurosci.* 2022

## Data Availability

The data are available upon request from the corresponding author.

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
