# Peer review of "Extracellular Vesicles in Inner Ear Therapies—Pathophysiological, Manufacturing, and Clinical Considerations"

_jcm, 2022, doi:10.3390/jcm11247455_

Round 1

Reviewer 1 Report

The authors wrote a narrative review on the potential uses of extracellular vesicles (EVs) in neurotology. This is a well-written and a needed review that will contribute to the search for otologic disease therapeutics. Below are some comments.

1. The authors gave a lot of literature on the inflammation and inflammatory diseases of the inner ear which should reflect in the title of the paper. In addition, the paper expanded on the application of EVs as therapeutic agents in the inner ear with not so many details on neurotology as suggested by the title. The authors should consider rephrasing the title. 

2. Lines 44-45. The prevalence of 1/6 hearing loss in the European population looks miss leading. The reference (“hear-it.org,”) may not be a good resource to determine the prevalence of hearing loss. Below is the current estimation from the website and it is clear that these numbers are arbitrary. Please consider reviewing the prevalence.

When asked, around one in ten will say they have a hearing loss, according to several surveys (https://www.hear-it.org/)

3. I suggest that the authors provide a table to summarize the studies that demonstrated that EVs can be used to protect the ear against hearing loss and the active/major cargo components of these EVs. I believe this will be very useful to the readers.

4. What are the current standardization issues with respect to EVs as therapeutics? Can the synthesis of EVs be standardized such that their cargo content is consistent in a way that mitigates batch-to-batch variation? Can the authors propose some standardization guidelines?

5. Can the authors clarify if there are clinical trials on the application of EVs in otologic diseases? It would be helpful to discuss a few clinical trials on the use of EVs in otologic diseases. 

Author Response

  1. The authors gave a lot of literature on the inflammation and inflammatory diseases of the inner ear which should reflect in the title of the paper. In addition, the paper expanded on the application of EVs as therapeutic agents in the inner ear with not so many details on neurotology as suggested by the title. The authors should consider rephrasing the title.
    1. Answer: Thank you for you comment. You are right. In order to better reflect the content of the manuscript, we have changed the title of the manuscript to: Extracellular Vesicles in Inner Ear Therapies – Pathophysiological, Manufacturing and Clinical Considerations

2. Lines 44-45. The prevalence of 1/6 hearing loss in the European population looks miss leading. The reference (“hear-it.org,”) may not be a good resource to determine the prevalence of hearing loss. Below is the current estimation from the website and it is clear that these numbers are arbitrary. Please consider reviewing the prevalence.

2. Answer: Thank you for your comment. We have changed the information. We used the WHO as a reference and corrected the incidence to 1 in 20, which equals 5 %.

3. I suggest that the authors provide a table to summarize the studies that demonstrated that EVs can be used to protect the ear against hearing loss and the active/major cargo components of these EVs. I believe this will be very useful to the readers.

3. Answer: Thank you for your comment. We have included a table summarising the studies showing inner ear protection of EVs; since many of the studies do not prove the active compounds, we omitted this information from the Table. 

4. What are the current standardization issues with respect to EVs as therapeutics? Can the synthesis of EVs be standardized such that their cargo content is consistent in a way that mitigates batch-to-batch variation? Can the authors propose some standardization guidelines?

4. Answer: Thank you for pointing out this very interesting and important issue. Standardisation issues have been discussed in detail other publications for example in the ISEV guidelines, to which we refer in our review. Currently, a revision of the guidelines is planned (personal communication during the recent meeting in Salzburg: Small New World) to specifically point out both the necessity and the difficulties in developing and defining standardisation guidelines with the help of hundreds of EV experts. Thus, we feel that for us it would be an impossible task to include such (to our knowledge not yet existing) standardisation guidelines to our Review articles. 

5. Can the authors clarify if there are clinical trials on the application of EVs in otologic diseases? It would be helpful to discuss a few clinical trials on the use of EVs in otologic diseases. 

5. Answer: Thank you for your comment. Currently, there are no clinical trials investigating the application of EVs in otological diseases. We have added this information in line 416 of our manuscript.

Reviewer 2 Report

Well written review on the causes of hearing loss, the role of inflammation, and the scientific basis for the use of exosomes.

This article outlines a significant public health issue, hearing loss, the state-of-the-art in discovery and covers the greatest hurdle to advancement, delivery.  This group has done significant work on exosomes from the inner ear.  They provide a detailed yet concise discussion of the potential for exosome delivery of therapeutics to the ear to combat inflammation.  The review presents relevant data available in the literature and will be useful to the interested reader.

Author Response

Well written review on the causes of hearing loss, the role of inflammation, and the scientific basis for the use of exosomes.

This article outlines a significant public health issue, hearing loss, the state-of-the-art in discovery and covers the greatest hurdle to advancement, delivery.  This group has done significant work on exosomes from the inner ear.  They provide a detailed yet concise discussion of the potential for exosome delivery of therapeutics to the ear to combat inflammation.  The review presents relevant data available in the literature and will be useful to the interested reader.

Answer: We thank the reviewer so much for his kind comments and specifically for the time and effort invested to review our manuscript.
